# High Levels of Low-Density Lipoproteins Correlate with Improved Survival in Patients with Squamous Cell Carcinoma of the Head and Neck

**DOI:** 10.3390/biomedicines9050506

**Published:** 2021-05-04

**Authors:** Torben Wilms, Linda Boldrup, Xiaolian Gu, Philip J. Coates, Nicola Sgaramella, Karin Nylander

**Affiliations:** 1Department of Medical Biosciences, Umea University, Umea, 901 87 Vasterbotten, Sweden; torben.wilms@umu.se (T.W.); linda.boldrup@umu.se (L.B.); sgaramellanicola12@gmail.com (N.S.); karin.nylander@umu.se (K.N.); 2Department of Clinical Sciences, Umea University, Umea, 901 87 Vasterbotten, Sweden; 3Research Centre for Applied Molecular Oncology, Masaryk Memorial Cancer Institute, Brno, 656 53 Southern Moravia, Czech Republic; philip.coates@mou.cz

**Keywords:** lipoprotein, SCCHN, prognosis

## Abstract

Circulating lipoproteins as risk factors or prognostic indicators for various cancers have been investigated previously; however, no clear consensus has been reached. In this study, we aimed at evaluating the impact of serum lipoproteins on the prognosis of patients with squamous cell carcinoma of the head and neck (SCCHN). Levels of total cholesterol, low-density lipoproteins (LDL), high-density lipoproteins (HDL), triglycerides and lipoprotein(a) were measured in serum samples from 106 patients and 28 healthy controls. We found that HDL was the only lipoprotein exhibiting a significant difference in concentration between healthy controls and patients (*p* = 0.012). Kaplan–Meier survival curves indicated that patients with high levels of total cholesterol or LDL had better overall survival than patients with normal levels (*p* = 0.028 and *p* = 0.007, respectively). Looking at patients without lipid medication (*n* = 89) and adjusting for the effects of TNM stage and weight change, multivariate Cox regression models indicated that LDL was an independent prognostic factor for both overall (*p* = 0.005) and disease-free survival (*p* = 0.013). In summary, our study revealed that high LDL level is beneficial for survival outcome in patients with SCCHN. Use of cholesterol-lowering medicines for prevention or management of SCCHN needs to be evaluated carefully.

## 1. Introduction

Squamous cell carcinoma of the head and neck (SCCHN) is a common tumor type and a significant cause of death worldwide [1]. The most common intraoral site for SCCHN is the tongue and squamous cell carcinoma of the oral tongue (SCCOT) is a serious public health problem with significant morbidity and mortality [2,3].

Several studies have shown that blood lipids are involved in the initiation and development of different types of cancer, including oral cancer [4]. Among blood lipids, triglycerides (TGs) and cholesterol are the most abundant, with cholesterol being essential for cell membrane biogenesis, proliferation and differentiation [5] and also being involved in the production of vitamin D and steroid hormones [6]. In humans, cholesterol is synthesized mainly by the liver, but is also provided in the diet. Low-density lipoproteins (LDL) transport cholesterol from the liver to cells, and high-density lipoproteins (HDL) transport cholesterol from cells to the liver [6].

It is well-known that chronically sick people have lower circulating levels of total cholesterol and that the levels of LDL and the LDL/HDL ratio are predictors of different diseases and outcomes [7,8,9,10].

A major topic of interest is the potential association between cholesterol and cancer. The exact role of cholesterol in the development of SCCHN is not clear and studies have shown contradictory results, where development of SCCHN is related to either a decrease or an increase in serum cholesterol, HDL and LDL [4,11,12,13]. A variant of LDL is lipoprotein(a), which is a risk factor for atherosclerosis and related diseases. Higher levels of lipoprotein(a) have been associated with poor prognosis in SCCHN [14].

Obesity, defined as body mass index (BMI) ≥ 30 kg/m^2^, has become a worldwide health problem and a risk factor for several malignancies [15]. Lipid alterations associated with obesity are decreased circulating HDL, and/or increased LDL, total cholesterol and TGs [16]. A recent study showed obesity to be an independent predictor of poor prognosis for patients with oral squamous cell carcinoma (OSCC) [17].

The aim of this study was to analyze blood samples from patients with SCCOT and other subtypes of SCCHN and map lipoprotein profiles and BMI in correlation to clinicopathological features to evaluate the potential of blood lipids as prognostic markers.

## 2. Materials and Methods

### 2.1. Patient Cohort

A cohort of 106 patients with clinically and histopathologically confirmed primary SCCHN were included. Of these, 28 had tumors in the oral mobile tongue (SCCOT), 9—on the floor of the mouth, 21—in gingivae, and 48—in the oropharynx (tonsils and/or base of the tongue). A group of 28 healthy volunteers without pre-cancerous lesions or any history of malignancies was included for control. Informed consent had been given by all the participants, and the study was approved by the local ethics committee (Dnr 08–003M). For SCCHN patients, a thorough clinical history including weight change judged as “No change”, “Increase of 1–5 kg”, “Decrease of 1–5 kg” and “Decrease of more than 5 kg” during the three months before diagnosis was taken [18]. For clinical data, see Table 1.

### 2.2. Blood Collection and Lipoprotein Analysis

Blood was collected from SCCHN patients in connection with diagnostic examination/surgical procedure before initiation of treatment. From both SCCHN patients and controls, peripheral blood (3 mL) was collected into vacutainers (SST™ II; cat. No. 368498; BD Biosciences, San Jose, CA, USA) using standardized venepuncture procedures. Tubes contained a serum separator, an acrylic-based gel forming a barrier between the clot and the serum after centrifugation, but no anticoagulant. After at least 30 min at room temperature, the tubes were centrifuged at 1300× g for 10 min at room temperature, followed by collection of the serum layer which was then stored at −80 °C until further use.

Levels of total cholesterol, HDL, LDL, TGs and lipoprotein(a) were analyzed at the accredited laboratory at Clinical Chemistry, Umeå University Hospital, NUS. Total cholesterol, HDL, TGs and lipoprotein(a) were measured directly, and the Friedewald equation was used to calculate LDL levels [19]. The unit for total cholesterol, HDL, LDL and TGs was mmol/L, for lipoprotein(a)—nmol/L.

### 2.3. Statistical Analysis

Comparison of lipoprotein levels between controls and patients was performed using the non-parametric Mann–Whitney U-test. Associations between categorized clinicopathological variables and categorized lipoprotein levels were determined by the chi-squared test. Associations between continuous variables were determined by the Spearman correlation analysis. The Kaplan–Meier method with log-rank test was used for survival analysis. The following cut-off levels were used for classification into high-/low-risk or normal: for cholesterol, 5.0 mmol/L; for LDL, 3.0 mmol/L; for HDL, 1.0 mmol/L for males and 1.2 mmol/L for females; for TGs, 1.7 mmol/L; for lipoprotein(a), 75 nmol/L [20,21]. Follow-up time for overall survival was calculated from time of diagnosis. Disease-free survival was defined as the time from completion of treatment to recurrence or death. In the Cox’s regression model, levels of cholesterol, LDL, age at diagnosis, BMI and weight change were treated as continuous variables. Gender, alcohol, smoking, HPV status, T stage, lymph node metastasis and TNM stage were categorical variables. For the multivariate Cox regression analysis, TNM stage and weight change were considered covariates. All statistical tests were conducted in IBM SPSS Statistics 25. A two-sided *p*-value < 0.05 was considered significant.

## 3. Results

### 3.1. Serum Lipid Levels in SCCHN Patients Compared to Controls

There was no significant difference between total cholesterol levels in SCCHN patients and controls (*p* = 0.208). Similarly, no significant difference in LDL levels was found between patients and controls (*p* = 0.554), whereas HDL levels were lower in patients (*p* = 0.012) (Figure 1A). As the Friedwald equation cannot be used for calculation of LDL levels if TG levels are too high (> 4.5 mmol/L), six samples (four SCCHN patients and two controls) lacked values for LDL. Since tumors in different sublocations of the head and neck area vary in characteristics [22,23], patients were divided according to sublocation of the tumor, exhibiting significant differences only for HDL in patients with gingival and oropharyngeal SCC (*p* = 0.003 and 0.013, respectively) (Figure 1B).

For analysis of TGs and lipoprotein(a), it is recommended to use serum samples collected at fasting stage. Our samples were taken before, at or after diagnostic examination/surgery, thus, the fasting stage was variable. All the patients were required to fast before surgery, while during surgery, they received an infusion drip containing glucose, and after surgery, the fasting status was not known. Dividing results into three different timepoints of collection showed a difference in levels of both TGs and lipoprotein(a) (*p* < 0.05, Figure 1C,D).

### 3.2. Clinicopathological Data in Correlation to Levels of Lipoproteins

When analyzing clinicopathological data in correlation to levels of lipoproteins, correlations between overall survival and cholesterol (*p* = 0.048), overall survival and LDL (*p* = 0.017) and disease-free survival and LDL (*p* = 0.030) were found (Table 2). For levels of cholesterol and BMI, we further analyzed correlations using the Spearman correlation analysis of non-categorized values. The results showed a positive correlation between LDL and BMI (correlation coefficient = 0.217, *p* = 0.029), whereas a negative correlation between HDL and BMI was observed (correlation coefficient = −0.380, *p* < 0.0001) (Appendix A).

It is known that carcinogens in tobacco can damage lipids and other cell membrane components [24]; however, in contrast to other studies, we observed no correlation between the lipid profile and smoking.

### 3.3. Levels of Lipoproteins in Correlation to Survival

Next, we analyzed the impact of serum lipoprotein levels on patient survival using the Kaplan–Meier method. SCCHN patients with high levels of total cholesterol exhibited longer overall survival than patients with normal levels (*p* = 0.028) (Figure 2A). As 17 patients were on anti-lipid medication, we next analyzed patients without lipid medication only (*n* = 89), and the results still showed high levels of cholesterol to be beneficial for survival (*p* = 0.017) (Figure 2B). Significant difference in overall survival was also observed when dividing patients into four groups according to both lipid medication status and cholesterol levels (Figure 2C, *p* < 0.0001). Patients without lipid medication but high cholesterol level had the best survival outcome. In general, overall survival in patients without lipid medication was better than in those with lipid medication (*p* = 0.007).

Levels of LDL also correlated to overall survival for SCCHN patients, where high levels of LDL were advantageous (*p* = 0.007) (Figure 2D). When selecting only patients without lipid medication (*n* = 85), high levels of LDL remained significantly correlated to longer survival in patients (*p* = 0.010) (Figure 2E). Dividing patients into four groups according to both lipid medication status and cholesterol levels, we found that patients without lipid medication but high LDL level had the best survival outcome (*p* < 0.0001) (Figure 2F).

HDL levels did not correlate with overall survival (*p* = 0.872) (Figure 2G). As levels of TGs and lipoprotein(a) were influenced by the fasting status, only the samples taken at fasting were analyzed. No significant differences in overall survival were observed for either of these two factors (Figure 2H,I).

We further analyzed the impact of BMI on overall survival. The group of overweight patients (defined as having a BMI of 25–30) exhibited better survival compared to normal-weight (BMI 18.5–24), obese (BMI > 30) and underweight patients (BMI < 18.5) (*p* < 0.001) (Figure 2J). The overall survival for oropharyngeal cancer in relation to the HPV status was also analyzed; however, no significant differences were found (*p* = 0.177, Appendix A).

The Kaplan–Meier survival analysis of disease-free survival according to levels of lipoproteins, BMI and HPV status also indicated that total cholesterol, LDL and BMI were significantly correlated with disease-free survival (data not shown).

Using the univariate Cox regression model for patients without lipid medication, impacts of total cholesterol, LDL, T stage, TNM stage and weight change on patient survival were observed (Table 3). Adjusting for the effects of TNM stage and weight change, the multivariate Cox regression indicated that LDL levels remained an independent prognostic factor for both overall (*p* = 0.005) and disease-free survival (*p* = 0.013), whereas total cholesterol was an independent prognostic factor only for overall survival (*p* = 0.033).

## 4. Discussion

The role of lipids in tumor development has been a focus of interest in many recent studies, but findings from different tumor types are often contradictive [25]. Previous studies have shown an association between blood cholesterol levels and different cancers [26] such as breast [27], colorectal [28,29] and lung cancer [30]. So far, there are only a few studies available on serum lipid profiles in SCCHN. In accordance with previous studies [4,11,30], we found levels of total cholesterol, LDL and HDL to be lower in SCCHN patients than in controls, although HDL was the only lipoprotein that was statistically significantly lower in the SCCHN patients in our study. Notably, 42% of the patients in our study had T4 tumors. Patients with tumors in advanced stages might have lower lipid levels due to food intake difficulties and malnutrition. However, no correlation between the T stage and lipid levels were observed in this study, excluding the possible link between tumor stages, malnutrition and low lipid levels in these patients. A reduction in HDL was observed in numerous previous studies and low HDL is believed to be an additional cancer predictor and is proposed to be caused by excess cholesterol utilization for membrane biogenesis by proliferating malignant cells [24].

LDL is composed of several proteins and lipids carrying cholesterol into peripheral tissues and also affecting the metabolism of fatty acids. Recent reports have indicated an emerging role of LDL in breast cancer, affecting cell proliferation and migration to facilitate disease progression [27].

In our group of SCCHN patients, high levels of LDL correlated with improved survival, whereas a previous study on SCCHN did not show a correlation between LDL and survival [14]. Apart from using different methods for detection of LDL, Li et al. excluded patients with a family history of lipidemia as well as obese persons, differences that could aid in explaining the different results achieved. In another study of 1081 patients with locoregionally advanced nasopharyngeal cancer, high LDL tended to be inferior for overall survival in comparison with low LDL [13]. Furthermore, a previous study of 601 patients with small-cell lung cancer showed lower LDL to be an independent prognostic factor for longer overall survival [31]. However, in patients with esophageal squamous cell carcinoma, low serum LDL levels were predictive factors for poor prognosis [32]. In ovarian cancer, longer overall survival has been observed for patients with normal compared to elevated levels of LDL [33]. However, it must be kept in mind that cholesterol is involved in formation of steroid hormones like estrogen and can thus have different effects in hormone-dependent cancer types (ovary, breast) compared to non-hormone-dependent cancers like SCCHN.

Cholesterol synthesis decreases in the liver with statin treatments which promote cholesterol uptake from plasma through upregulation of LDL receptors on the surface of hepatocytes, also increasing LDL degradation in the liver. The use of statins is now common and when excluding patients on statin medication (17 patients), we still obtained significantly better survival for patients with high levels of total cholesterol and LDL. As cholesterol levels were lower in SCCHN patients compared to healthy controls, it could be speculated that higher levels mimic the normal state.

Another interesting finding in our study was that the group of overweight, but not obese, patients (pre-treatment BMI of 25–30) exhibited better survival compared to normal weight, obese and underweight patients. This is in accordance with other studies that higher BMI is related to better survival and lower recurrence and distant metastasis rates in SCCHN [34,35]. Obesity, on the other hand, which is characterized by increased levels of LDL, cholesterol and TGs, as well as by decreased HDL [16], was previously identified as an independent adverse prognostic variable, where obese SCCOT patients had a five-fold increase in the risk of death compared to normal-weight patients [36].

Even if measuring a single pre-treatment LDL level may not adequately reflect the variance in lipoproteins over the clinical course of SCCHN, these data are among the first to examine LDL levels as a predictor of outcome in SCCHN. The results also suggest a mechanism by which cholesterol-altering drugs may be used to influence outcomes in the future. Based on the present results showing improved survival for patients with high LDL, it could also be speculated whether statin medication should be withdrawn for SCCHN patients being treated for their tumor.

## Figures and Tables

**Figure 1 biomedicines-09-00506-f001:**
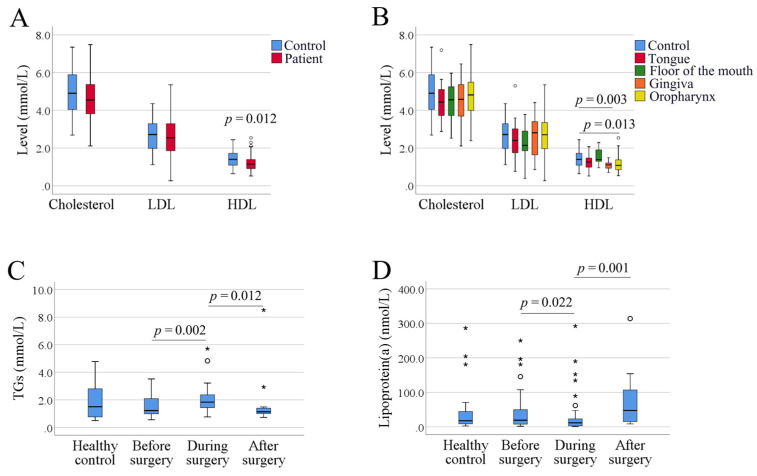
Box plots of serum lipoprotein levels in healthy controls and patients with SCCHN. (**A**) Levels of total cholesterol, low-density lipoproteins (LDL) and high-density lipoproteins (HDL). (**B**) Patients were divided into four subgroups according to tumor locations. (**C**) Levels of triglycerides (TGs) in healthy controls and patients with different timepoints of blood collection. (**D**) Levels of lipoprotein(a) in healthy controls and patients with different timepoints of blood collection. Small circles indicate outliers and asterisks indicate extreme outliers.

**Figure 2 biomedicines-09-00506-f002:**
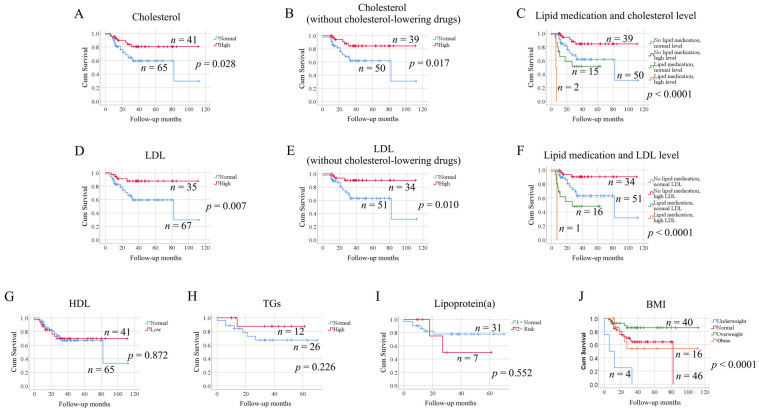
Kaplan–Meier curves showing the impact of lipoprotein levels on the overall survival of patients with SCCHN. (**A**) Comparison between cholesterol-high and cholesterol-normal patients. (**B**) The impact of cholesterol levels on the overall survival was investigated only for patients without cholesterol-lowering drugs. (**C**) Patients were grouped according to the lipid medication status and cholesterol levels. (**D**) Comparison between LDL-high and LDL-normal patients. (**E**) The impact of LDL levels on the overall survival was investigated only for patients without cholesterol-lowering drugs. (**F**) Patients were grouped according to the lipid medication status and LDL levels. (**G**) Comparison between HDL-low and HDL-normal patients. (**H**) Comparison between TGs-high and TGs-normal patients (only the samples taken at fasting were analyzed). (**I**) Comparison between lipoprotein(a)-risk and lipoprotein(a)-normal patients (only the samples taken at fasting were analyzed). (**J**). Comparison of patients with different BMI levels.

**Table 1 biomedicines-09-00506-t001:** Clinical features of patients with SCCHN.

Clinical Features		Number of Patients
Tumor Localization	Tongue	28
	Floor of the mouth	9
	Gingivae	21
	Oropharynx (tonsils and/or base of the tongue)	48
Age at Diagnosis	≤40 years	8
	41 to 65 years	50
	>65 years	48
Gender	Female	36
	Male	70
Smoking	Non-smoker	38
	Previous smoker	31
	Smoker	37
Alcohol	No	24
	Yes	82
HPV Status	Negative	67
	Positive	39
BMI	Underweight (<18.5)	4
	Normal (18.5 to 24)	46
	Overweight (25 to 30)	40
	Obese (>30)	16
Weight change	No change	64
	Increase of 1 to 5 kg	4
	Decrease of 1 to 5 kg	20
	Decrease > 5 kg	18
Cholesterol-lowering Medication	No	89
	Yes	17
T Stage	T1	22
	T2	35
	T3	5
	T4	44
Lymph Node Metastasis	No	50
	Yes	56
Distant Metastasis	No	104
	Yes	2
Clinical TNM Stage (8th edition)	I	35
	II	16
	III	19
	IV	1
	IVa	30
	IVb	4
	IVc	1
Treatment	Surgery alone	12
	Surgery with postoperative radiotherapy	27
	Surgery with postoperative radiotherapy and chemotherapy	2
	Radiotherapy alone	38
	Preoperative radiotherapy and surgery	15
	Radiotherapy in combination with chemotherapy or pharmacotherapy	12
Status	Alive	76
	Dead	30
Total		106

**Table 2 biomedicines-09-00506-t002:** Correlations between clinical variables and levels of cholesterol/LDL/HDL.

Clinical Variables		Total Cholesterol			LDL			HDL		
		Normal	High	*p*-Value	Normal	High	*p*-Value	Normal	Low	*p*-Value
Age at diagnosis (years)	20–40	5	3	0.066	6	2	0.092	3	5	0.302
	41–65	25	25		25	21		33	17	
	> 65	35	13		36	12		29	19	
Gender	Female	23	13	0.834	26	10	0.384	24	12	0.528
	Male	42	28		41	25		41	29	
Alcohol	Yes	49	33	0.637	52	26	0.807	51	31	0.813
	No	16	8		15	9		14	10	
Smoking	Smoker	19	18	0.244	23	12	0.233	25	12	0.558
	Previous smoker	22	9		23	7		17	14	
	Non-smoker	24	14		21	16		23	15	
BMI	Underweight	3	1	0.374	4	0	0.351	3	1	0.093
	Normal	32	14		32	14		34	12	
	Overweight	21	19		23	16		20	20	
	Obese	9	7		8	5		8	8	
Cholesterol-lowering medication	No	50	39	0.014	51	34	0.005	54	35	1.000
	Yes	15	2		16	1		11	6	
T stage	T1, T2	36	21	0.694	39	16	0.296	38	19	0.237
	T3, T4	29	20		28	19		27	22	
Lymph node metastasis	Negative	28	22	0.322	29	19	0.305	32	18	0.690
	Positive	37	19		38	16		33	23	
Clinical TNM stage	I, II	33	18	0.552	34	16	0.680	34	17	0.321
	III, IV	32	23		33	19		31	24	
HPV status	Negative	42	25	0.836	44	20	0.518	46	21	0.062
	Positive	23	16		23	15		19	20	
Weight change (kg)	No change	37	27	0.454	40	23	0.144	41	23	0.121
	Increase of 1–5	2	2		2	1		2	2	
	Decrease of 1–5	12	8		10	9		15	5	
	Decrease > 5	14	4		15	2		7	11	
Overall survival status	Alive	42	34	0.048	44	31	0.017	46	30	0.828
	Dead	23	7		23	4		19	11	
Disease-free status	Disease-free	44	34	0.113	46	31	0.030	50	28	0.370
	With disease	21	7		21	4		15	13	

**Table 3 biomedicines-09-00506-t003:** Cox regression analysis for patient survival (patients without lipid medication).

Group			Overall Survival				Disease-Free Survival			
			*p*-Value	Hazard Ratio	95% CI for Hazard Ratio		*p*-Value	Hazard Ratio	95% CI for Hazard Ratio	
					Lower	Upper			Lower	Upper
Univariate Cox regression analysis	Cholesterol		0.021	0.599	0.387	0.926	0.068	0.676	0.444	1.029
	LDL		0.006	0.514	0.319	0.827	0.015	0.551	0.341	0.890
	Age		0.113	1.031	0.993	1.071	0.115	1.030	0.993	1.068
	Gender		0.559	0.763	0.308	1.891	0.355	0.664	0.278	1.583
	Alcohol		0.971	0.980	0.327	2.935	0.817	1.126	0.412	3.077
	Smoking		0.730	1.087	0.677	1.744	0.964	1.011	0.628	1.627
	BMI		0.080	0.901	0.801	1.013	0.073	0.907	0.816	1.009
	HPV status		0.568	0.757	0.291	1.971	0.300	0.606	0.235	1.563
	T stage		0.065	1.460	0.977	2.183	0.046	1.489	1.007	2.201
	Lymph node metastasis		0.064	2.374	0.952	5.918	0.192	1.793	0.746	4.312
	TNM stage		0.011	1.376	1.075	1.762	0.002	1.510	1.166	1.956
	Weight change		0.011	0.903	0.834	0.977	0.024	0.915	0.846	0.988
Multivariate Cox regression analysis	Independent predictors	Cholesterol	0.033	0.599	0.374	0.959	0.105	0.691	0.442	1.080
		TNM stage	0.021	1.355	1.047	1.755	0.004	1.488	1.138	1.944
		Weight change	0.088	0.931	0.858	1.011	0.232	0.951	0.875	1.033
	Independent predictors	LDL	0.005	0.451	0.259	0.787	0.013	0.501	0.290	0.867
		TNM stage	0.034	1.357	1.024	1.800	0.006	1.494	1.122	1.991
		Weight change	0.046	0.916	0.841	0.998	0.154	0.941	0.865	1.023

## Data Availability

The data that support the findings of this study are available from the corresponding author.

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
