# Peer review of "High Levels of Low-Density Lipoproteins Correlate with Improved Survival in Patients with Squamous Cell Carcinoma of the Head and Neck"

_biomedicines, 2021, doi:10.3390/biomedicines9050506_

Round 1

Reviewer 1 Report

#1. line 26 – 27 “Use of cholesterol lowering medicine for prevention or management of SCCHN need to be evaluated carefully” >> please report the status of using cholesterol lowering medicine in table 1 and included this “the status of using cholesterol lowering medicine” in survival analyses.

#2. line 65- 66 “28 healthy volunteers without any history of SCCHN or other malignancies was included as controls.” >> do these 28 volunteers had pre-cancer lesions?

#3. line 116 “Figure 2. Box-plots of serum lipoprotein levels in healthy controls and patients with SCCHN” >> or “Figure 1. Box-plots of serum lipoprotein levels in healthy controls and patients with SCCHN” ?

#4. table 1: distal metastases n =2 but stage IVc n=1 >> please double check; also all clinical staging? Or mixed clinical and pathological staging?

Author Response

#1. line 26 – 27 “Use of cholesterol lowering medicine for prevention or management of SCCHN need to be evaluated carefully” >> please report the status of using cholesterol lowering medicine in table 1 and included this “the status of using cholesterol lowering medicine” in survival analyses.

Reply: Lipid medication status is now summarized in Table 1 on page 3. Correlations between status of lipid medication and levels of cholesterol/LDL/HDL are also added in Table 2, page 6.

Kaplan-Meier overall survival analysis indicated that patients without lipid medication (n = 89) had better survival than patients with lipid medication (n = 17). At line 147 on page 6, the following information is added: Significant difference in overall survival was also seen when dividing patients into 4 groups according to both lipid medication status and cholesterol levels (Figure 2C, p < 0.0001). Patients without lipid medication but high cholesterol level had the best survival outcome. In general, overall survival in patients without lipid medication was better than for those with lipid medication (p = 0.007).

Due to the significant association between lipid medication and patient survival, we now perform cox regression survival analysis only for patients without lipid medication. As shown on page 8, the results in Table 3 are updated.

#2. line 65- 66 “28 healthy volunteers without any history of SCCHN or other malignancies was included as controls.” >> do these 28 volunteers had pre-cancer lesions?

Reply: No pre-cancer lesions were reported in the healthy volunteers. This statement is now added to the revised manuscript (line 66, page 2).

#3. line 116 “Figure 2. Box-plots of serum lipoprotein levels in healthy controls and patients with SCCHN” >> or “Figure 1. Box-plots of serum lipoprotein levels in healthy controls and patients with SCCHN” ?

Reply: Sorry for the mistake! It should be Figure 1. This is now corrected, as shown at line 117, page 5.

#4. table 1: distal metastases n =2 but stage IVc n=1 >> please double check; also all clinical staging? Or mixed clinical and pathological staging?

Reply: Sorry for the mistake! For one patient with oropharynx HPV-positive T4N1M1 tumor, the clinical stage should be IV. This is now corrected, as shown in Table 1 on page 3.

Reviewer 2 Report

The authors measured levels of total cholesterol, LDL, HDH, triglyceride, and lipoproteins in the serum of HNSCC patients and found that patients with high total cholesterol or LDL had a longer overall survival than patients with normal levels. They concluded that high LDL levels were beneficial for survival outcomes in HNSCC patients and that the use of cholesterol-lowering drugs should be carefully evaluated.

The study appears to be the first to show the relationship between high LDL and improved survival in patients with HNSCC and is valuable in determining prognostic predictors in patients with HNSCC.

There are several points to reconsider, as described below.

The authors cited articles supporting the relationship between low LDL levels and longer overall survival in small-cell lung cancer, ovarian cancer, and breast cancer. Regarding HNSCC, Yao et al. (Cell Physiol Biochem 2018, 48, 285) reported that in locally advanced nasopharyngeal cancer, high LDL tended to be inferior in OS compared to low LDL. This paper can be added to the references.

Meanwhile, Chen et al. (Oncotarget 2017, 8, 41605) reported that low total and LDL cholesterol was associated with shorter disease-free survival and overall survival in esophageal SCC patients with esophagectomy. In the present study, patients with low LDL patients in normal level group may have a poorer prognosis. This may contribute to a better prognosis for patients with high cholesterol and LDL levels.

Low levels of total cholesterol and LDL have been repeatedly reported in HNSCC patients (ref 4,11, and 30). HNSCC patients, especially those in advanced stages, can be malnourished. In the present study, 44/106 (42 %) of patients have T4 tumors, which can lead to food intake difficulties and malnutrition. The discussion will require these considerations.

The authors stated that based on the results showing improved survival for patients with high LDL, it could be speculated whether stain medication should be withdrawn for HNSCC patients being treated for their tumor. However, it is unclear whether the observation period is sufficient to determine the effect of stain on overall survival.

Tables 1 and 2: Hpv would be HPV. Tnm would be TNM.

Author Response

The authors measured levels of total cholesterol, LDL, HDH, triglyceride, and lipoproteins in the serum of HNSCC patients and found that patients with high total cholesterol or LDL had a longer overall survival than patients with normal levels. They concluded that high LDL levels were beneficial for survival outcomes in HNSCC patients and that the use of cholesterol-lowering drugs should be carefully evaluated.

The study appears to be the first to show the relationship between high LDL and improved survival in patients with HNSCC and is valuable in determining prognostic predictors in patients with HNSCC.

There are several points to reconsider, as described below.

The authors cited articles supporting the relationship between low LDL levels and longer overall survival in small-cell lung cancer, ovarian cancer, and breast cancer. Regarding HNSCC, Yao et al. (Cell Physiol Biochem 2018, 48, 285) reported that in locally advanced nasopharyngeal cancer, high LDL tended to be inferior in OS compared to low LDL. This paper can be added to the references.

Reply: Thank you for your suggestion. This reference is now added to the revised manuscript (line 50 on page 2, and line 216 on page 9).

Meanwhile, Chen et al. (Oncotarget 2017, 8, 41605) reported that low total and LDL cholesterol was associated with shorter disease-free survival and overall survival in esophageal SCC patients with esophagectomy. In the present study, patients with low LDL patients in normal level group may have a poorer prognosis. This may contribute to a better prognosis for patients with high cholesterol and LDL levels.

Reply: Yes, this finding is consistent with ours. This reference is now added to line 220 on page 9.

Low levels of total cholesterol and LDL have been repeatedly reported in HNSCC patients (ref 4,11, and 30). HNSCC patients, especially those in advanced stages, can be malnourished. In the present study, 44/106 (42 %) of patients have T4 tumors, which can lead to food intake difficulties and malnutrition. The discussion will require these considerations.

Reply: Yes, patients with tumors in advanced stages might have lower lipid levels due to food intake difficulties and malnutrition. However, as shown in Table 2, there is no correlation between T stage and levels of cholesterol/LDL/HDL. This discussion is now added to the revised manuscript (line 200, page 9).

The authors stated that based on the results showing improved survival for patients with high LDL, it could be speculated whether stain medication should be withdrawn for HNSCC patients being treated for their tumor. However, it is unclear whether the observation period is sufficient to determine the effect of stain on overall survival.

Reply: This is an interesting question, however, the number of patients on lipid medication in this study is fairly low, 17, and definite conclusions will thus be hard to draw. Also, just considering lipid medication in overall survival is not enough, as other factors, not included in this study, such as dietary habits, also must be taken into consideration.

Tables 1 and 2: Hpv would be HPV. Tnm would be TNM.

Reply: This is now corrected, as shown on page 3 and 6.

Round 2

Reviewer 1 Report

# table 1: clinical staging: Could it be I/II/III/Iva/IVb/IVc = 35/16/19/30/4/2? To the reviewer’s understanding, stage IVc HNSCC was usually assigned for those with distant metastases [n=2 in table 1]

Author Response

# table 1: clinical staging: Could it be I/II/III/Iva/IVb/IVc = 35/16/19/30/4/2? To the reviewer’s understanding, stage IVc HNSCC was usually assigned for those with distant metastases [n=2 in table 1] Reply: Thank you for this question. According to the 8th Edition Staging Manual, for patients with HPV-associated oropharyngeal cancer, stage IV is reserved for patients with distant metastatic disease, this is different to non-HPV-associated cancers in which stage IV is subdivided into IVa, IVb and IVc.